

# The recent expansion of Fox Sparrow (*Passerella iliaca iliaca*) breeding range into the northeastern United States

John D. Lloyd

Vermont Center for Ecostudies, Norwich, VT, USA

## ABSTRACT

The breeding range of the Eastern Fox Sparrow (*Passerella iliaca iliaca*) is generally recognized as comprising the boreal forest of Canada. However, recent observations suggest that the species is present during the summer months throughout much of the northeastern US, unexpected for a species characterized as a passage migrant in the region. To clarify, I conducted a literature review to document the historical status of the species in the northeastern US and then analyzed observations submitted to eBird to describe its recent and current status in the region. Historical accounts consistently identify Fox Sparrow as a passage migrant through the region during early spring and late fall. Beginning in the early 1980s, observers began noting regular extralimital records of Fox Sparrow in northern Maine. A single nest was discovered in the state in 1983, and another in northern New Hampshire in 1997. Despite the paucity of breeding records, observations submitted to eBird suggest that the southern limit of the breeding range of Fox Sparrow has expanded rapidly to the south and west in recent years. The proportion of complete checklists submitted to eBird that contained at least one observation of Fox Sparrow grew at an annual rate of 18% from 2003–2016 and was independent of observer effort. Fox Sparrow now occurs regularly on mountaintops and in young stands of spruce (*Picea* spp.) and balsam fir (*Abies balsamea*) during the breeding season throughout northern and western Maine and northern New Hampshire, with occasional records from the Green Mountains of Vermont and the Adirondack Mountains of New York. The cause of this rapid expansion of its breeding range is unknown, but may be related to an increase in the amount of young conifer forest in the northeastern US created by commercial timber harvest.

## INTRODUCTION

Fox Sparrow (*Passerella iliaca*) is a polytypic species that breeds in montane and boreal forest across western and northern North America. Eastern Fox Sparrow, (*P. i. iliaca*), part of the Red Fox Sparrow subspecies group (along with *P. i. zaboria*), nests in early-successional coniferous or mixed forests; shrubby thickets along waterways and wetlands; and stunted conifer forests on mountaintops or cool coastlines from Manitoba eastward to the Maritime Provinces of Canada (*Bisson & Limoges, 1996*; *McLaren, 2007*; *Stewart, 2015*; *Artuso, 2018*).

Corresponding author
John D. Lloyd,
jlloyd@vtecostudies.org,
5355693@gmail.com

Most general references identify the breeding range of Eastern Fox Sparrow as extending into the eastern United States only in the northernmost part of Maine (*Rising, 1996*; *Sibley, 2000*; *Weckstein, Kroodsma & Faucett, 2002*). However, anecdotal reports from birders and observations submitted to eBird (http://www.ebird.org), a web platform that documents bird distributions using observations submitted by citizen-scientists (*Sullivan et al., 2009*), suggest that Fox Sparrow now occurs regularly during the breeding season throughout northern and western Maine and as far south and west as the mountains of central and northern New Hampshire. This would constitute a fairly rapid southward expansion of the species' breeding range, on the order of several hundred kilometers, yet this phenomenon has remained undescribed in the ornithological literature. Here, in an effort to address this gap and to clarify the status of Fox Sparrow as a breeding species in the northeastern US, I review historical and current literature describing the distribution of Fox Sparrow in this region and describe temporal changes in occurrence using data submitted to eBird by citizen scientists.

## METHODS

I began by reviewing general summaries of bird distribution for the two northeastern states that appear to represent the leading edge of the southward expansion of Fox Sparrow breeding range: Maine, the bird life of which still has only a single definitive reference (*Palmer, 1949*), and New Hampshire, which has both an historical account of bird distributions (*Allen, 1903*) and an authoritative recent update (*Keith & Fox, 2013*). I also consulted two older accounts of the birdlife of New England (*Samuels, 1875*; *Forbush, 1929*). These accounts formed the basis for understanding the historic distribution of Fox Sparrow in Maine and New Hampshire.

I supplemented these historical accounts, and located more recent references to the species' distribution, by searching Google Scholar with the string "("fox sparrow" OR "passerella iliaca") AND ("New Hampshire" OR Maine)". I also manually searched for records of Fox Sparrow in regional reports for the northeastern US and southeastern Canada that appeared in American Birds and National Audubon Society Field Notes, published by the National Audubon Society, and North American Birds, published by the American Birding Association, from 1980–2006. These journals, a continuous series despite the changing titles and change in publishers, provide quarterly reviews of notable bird observations submitted from across North America, which are compiled and vetted by regional experts. American Birds covers the period 1980–1994, National Audubon Society Field Notes the period from 1994–1999, and North American Birds the period from 1999–2006. For records specific to New York, I manually searched the online archives (covering 2000–2016) of The Kingbird, the journal of the New York State Ornithological Association. Finally, I searched breeding bird atlases for Maine, New Hampshire, Vermont, and New York for any information pertaining to the presence of Fox Sparrows.

To describe and quantify recent changes in the incidence of Fox Sparrow reports during the breeding season, I used data submitted to eBird. To describe the current distribution, I used the July 2018 version of the eBird Basic Dataset, which includes both incidental

(i.e., potentially incomplete checklists without associated information on observer effort) and complete-checklist records (i.e., checklists that include all species observed and that have an accompanying measure of observer effort).

   To quantify temporal changes in the incidence of Fox Sparrow observations, I began by downloading the 2016 version of the eBird Reference Dataset (*Fink et al., 2017*), which includes only complete checklists and is zero-filled such that Fox Sparrows were treated as absent from any checklist without a positive record of the species. This download consisted of 14,728,627 checklists. I then filtered this dataset to include only checklists from June and July, which should eliminate nearly all records of migrant birds, as Fox Sparrow migration through the region occurs very early in April (*Weckstein, Kroodsma & Faucett, 2002*). The mean arrival date on the breeding grounds in Newfoundland, Canada, for example, was 9 April over a six-year period from 1973–1978 (*Threlfall & Blacquiere, 1982*). Regionally, the species is also a late fall migrant, with southward movements typically not beginning until mid- to late September (*Weckstein, Kroodsma & Faucett, 2002*). I further reduced the dataset by including only checklists from counties in the northern halves of Maine (Aroostook, Penobscot, Piscataquis, Somerset, Franklin, and Oxford) and New Hampshire (Coos and Grafton). I limited the analysis to this extent because it included nearly all of the potentially suitable habitat in both states, namely stunted krummholz forest on high mountaintops and young stands of balsam fir (*Abies balsamea*) and spruce (*Picea* spp.) regenerating after harvest in the region's extensive commercial forestland. Next, I filtered out any checklists associated with survey protocols not compatible with detection of Fox Sparrows during the breeding season, in particular checklists with count types identified as "Yard Count", "Loon Watch", "My Yard eBird", and "Pelagic" (although the former should be unnecessary given the geographic filters applied, some checklists in this filtered set were flagged as pelagic counts). I removed these checklists because, in surveying water bodies or residential areas, they cover environments impossible or unlikely to support breeding Fox Sparrows, although they accounted for only a small number of filtered records (1, 3, 6, and 13 checklists, respectively). Finally, I removed duplicate checklists generated by multiple observers birding together, first verifying that the number of Fox Sparrows recorded on each checklist matched. In 3 cases, Fox Sparrows were observed by only 1 member of a birding group (in all 3 cases, on the primary checklist) and in these cases I retained the checklist that included the observation of Fox Sparrow, the logic being that the bird was present even if only recorded on one of the group checklists. After applying these filters, I was left with 74,881 checklists, to which all subsequent analyses were applied (data available at https://doi.org/10.6084/m9.figshare.7283144.v1). I estimated the elevation for each complete checklist containing an observation of Fox Sparrows using a 30-m resolution digital-elevation model (*Farr et al., 2007*).

   Analyzing the regional occurrence of Fox Sparrows in eBird checklists posed two problems. First, because of the species' rarity and its localized distribution in the region, many checklists were submitted from the same locale on approximately the same date and presumably recorded the presence of the same individual or individuals. For example, when an individual was located on an accessible mountaintop, many birders would actively seek out that individual and submit a checklist recording its presence. As these are not

independent observations, this phenomenon would potentially lead to an overestimate of the frequency with which the species is observed in the region. At the same time, the localized regional distribution of Fox Sparrows resulted in an exceedingly small proportion of checklists reporting the species.

To address both issues, rather than use the raw proportion of checklists with Fox Sparrow, I instead used as a dependent variable the proportion of 10 km$^2$ grid cells within the region that had at least one complete checklist containing Fox Sparrow. To accomplish this, I first overlaid a 10 km$^2$ grid atop the study area within the QGIS version 2.18 Geographic Information System (*QGIS Development Team, 2016*) and then determined whether each grid cell contained at least one complete checklist with a record of Fox Sparrow for each year. Doing so avoided the problem of treating multiple observations of the same bird as independent and also yielded clearer insight into large-scale changes in the occurrence of Fox Sparrow. The choice of the grid size was essentially subjective, but reflected a trade-off between cells that were too large to accurately quantify changes in distribution over time and cells that were too small to accurately record the location where Fox Sparrows were detected during traveling counts submitted to eBird (i.e., situations where an observer recorded birds in multiple grid cells on a single checklist; the median length of a traveling checklist used in this analysis was 3.3 km and the mean length was 6.2 km). To test the sensitivity of the results to the choice of a size for the grid cells, I repeated the analysis described below using a 5 km$^2$ grid and found that the results were nearly identical (supplemental results of this analysis are available at https://doi.org/10.6084/m9.figshare.7300031.v1).

I analyzed changes in the proportion of grid cells occupied by Fox Sparrow over time using a generalized linear model, with the conditional distribution of the response variable assumed to be binomial (i.e., a grid cell did or did not contain a record of Fox Sparrow in that year). I included year as the predictor variable in the model. Any change in the proportion of cells with Fox Sparrow records could be due in part to increased observer effort over time, so I analyzed a second model that included both year and a measure of observer effort, which I quantified as the summed value of observation hours spent on each complete checklist in the grid cell. I used analysis of deviance to choose the preferred model for inference. I tested for overdispersion via a chi-squared test of the ratio of the squared sum of Pearson residuals to the residual degrees of freedom (*Venables & Ripley, 2002*). None of the models showed evidence of overdispersion (i.e., $\chi^2 P < 0.05$), and as such I made no adjustments.

Although the measure of observer hours spent birding can help address the confounding effect of effort on the apparent frequency with which a rare species like Fox Sparrow is detected, it does not address changes in observer behavior that may increase the likelihood of encountering the species. For example, over time, birders may have become more adventurous in their explorations and more likely to visit the locales—remote mountaintops or commercial forestlands with limited access—potentially inhabited by Fox Sparrows. To address this potential source of confounding, I repeated the above analysis for two other species that occupy similar habitat as Fox Sparrows: Bicknell's Thrush (*Catharus bicknelli*) and Blackpoll Warbler (*Setophaga striata*). Bicknell's Thrush populations have likely been declining slowly in the region (*Lambert et al., 2008*; *King et al., 2008*) in recent decades,

whereas Blackpoll Warblers have shown no significant trend in numbers (*King et al., 2008*), so any increase in the proportion of grid cells occupied by these species in the study area might reflect increased observer activity in habitat suitable for Fox Sparrows.

All analyses were conducted in R version 3.4.4 (*R Core Team, 2018*).

## RESULTS

### Historical distribution of Fox Sparrows in the northeastern United States

Early reviews of the region's avifauna were unequivocal in describing Fox Sparrow as a passage migrant. *Palmer (1949)* identified the species as transient in Maine; *Allen (1903)* as a rather common migrant in New Hampshire. *Keith & Fox (2013)* reached a similar conclusion in their comprehensive review of historical and recent records from New Hampshire, describing both the historic (1875–1950) and recent status as "transient" in spring and fall (while also noting a single, recent breeding record; see below). The regional summaries in both *Samuels (1875)* and *Forbush (1929)* classified the species as a spring and fall migrant throughout New England. None of these references indicated that Fox Sparrows were present historically in the region outside of early spring and late fall.

### Recent distribution of Fox Sparrows in the northeastern United States

Several checklists submitted as historical records to eBird reported Fox Sparrows in far northern Maine near Madawaska during the breeding season (47.36°N, 68.33°W) as early as 1978. The first strong indication that Fox Sparrows were breeding in the northeastern US came in 1981, documented in an eBird checklist submitted as an historical record in 2017. Three observers (Jeff Cherry, Jim Eckler, and Lynn Sheldon), working on a study of the effects on birds of spraying pesticides to control an outbreak of eastern spruce budworm (*Choristoneura fumiferana*), located a singing Fox Sparrow in northern Somerset County, Maine (46.36°N, 70.06°W) on several occasions between 21 June and 3 July. As described in the comments associated with the checklist, this bird was reported to have responded aggressively to a broadcast recording of a Fox Sparrow song. Confirmation of nesting was obtained further north (47.35°N, 68.28°W), several kilometers south of the US-Canada border in Aroostook County, Maine, in 1983 by Peter Vickery during Maine's first breeding bird atlas (*Adamus, 1988*).

Outside of these scattered records, evidence of a widespread southward range expansion first began emerging in the mid-1980s in southern Quebec. In 1985 and again in 1986, birders observed singing Fox Sparrows well south of their breeding range in southeast Quebec near the border of Maine (*Yank & Aubry, 1986*; *Yank & Aubry, 1985*). In 1988, four Fox Sparrows in two different locations in southern Quebec were also noteworthy for being "well s. of their usual summer range" (*Gosselin, Yank & Aubry, 1988*). In 1993, three singing Fox Sparrows were observed in "suitable nesting habitat in n. Maine during late June" (*Petersen, 1993*). The species was still uncommon enough to warrant attention in 1999 and 2000, when notable records of singing birds were obtained in central Maine near Mount Katahdin (*Petersen, 1999*; *Petersen, 2000*). Meanwhile, in northern New Hampshire,

a Fox Sparrow was discovered singing in 1996, with nesting confirmed at the same location (45.25°N, 71.21°W) in 1997 (*Keith & Fox, 2013*). This was the first confirmed nesting attempt by Fox Sparrow in New Hampshire, none having been found during the state's breeding bird atlas, conducted from 1981–1986 (*Foss, 1994*). The sole record of Fox Sparrow in New York came in 2012, with an individual observed on Whiteface Mountain (44.3659°N, 73.9026°W) between 14 June and 11 July (*McCormack, 2012*). Fox Sparrows were also absent from statewide breeding-bird atlases conducted in New York from 1980–1985 (*Andrle & Carroll, 1988*) and 2000–2005 (*McGowan & Corwin, 2008*), and in Vermont from 1976–1981 (*Laughlin & Kibbe, 1985*) and 2003–2007 (*Renfrew, 2013*).

The breeding distribution of Fox Sparrows at the time when the southward expansion began, based on eBird records, extended roughly as far south as the Gaspe Peninsula, northern New Brunswick, and Nova Scotia (Fig. 1). Based on all eBird records submitted through July 2018, including incidental observations or otherwise incomplete checklists, the current distribution of Fox Sparrow during June and July in the northeastern US includes northern Maine, the mountains of central and western Maine, northern New Hampshire, and the White Mountains of central New Hampshire (Fig. 2). In addition to the lone record from the Adirondacks, the species also has been observed several times in June and July in the central Green Mountains of Vermont. Including these isolated observations, Fox Sparrow now occurs as far south as 44°N in the high peaks of the White Mountains and as far west as 74°W in the Adirondack Mountains, or roughly 3° further south than it did 30 years earlier.

The single record from southern Maine (44.74°N, 68.73°W), in July 2016, is of unknown significance. Its location, at a frequently birded nature preserve near the city of Bangor with no other breeding season records of Fox Sparrow, suggests that bird observed was not on a breeding territory. Based on comments in the associated checklists, the eBird records from Vermont stem from surveys conducted by citizen-scientists working on the Vermont Center for Ecostudies' Mountain Birdwatch Program, an annual, trail-based survey of birds breeding at high elevations in the northeastern US. The detections occurred on 1 survey route in 2011, 1 in 2012, and 3 different routes in 2016.

## Temporal changes in the frequency of Fox Sparrow detections in the northeastern United States

Fox Sparrows and Blackpoll Warbler co-occurred on eBird checklists frequently, with 47.4% of checklists reporting Fox Sparrow also reporting Blackpoll Warbler, and 34.5% of checklists reporting Blackpoll Warbler also reporting Fox Sparrow. Fox Sparrows co-occured less often with Bicknell's Thrush (11.5% of checklists with Fox Sparrow also recorded Bicknell's Thrush), presumably reflecting the broader range of forest types—especially those at lower elevations—used by Fox Sparrow. Of the checklists reporting Bicknell's Thrush, 37.8% also reported Fox Sparrow, indicating the propensity of Fox Sparrows to occur in krummholz forests.

The proportion of grid cells containing complete eBird checklists that noted the presence of Fox Sparrow increased from 2003–2016 ($b_{year} = 0.18$, 95% CI [0.097–0.275]; $P < 0.001$; Fig. 3), ranging from a low of 0% in 2003 (0/28 cells), 2004 (0/29), 2006 (0/32), and 2007

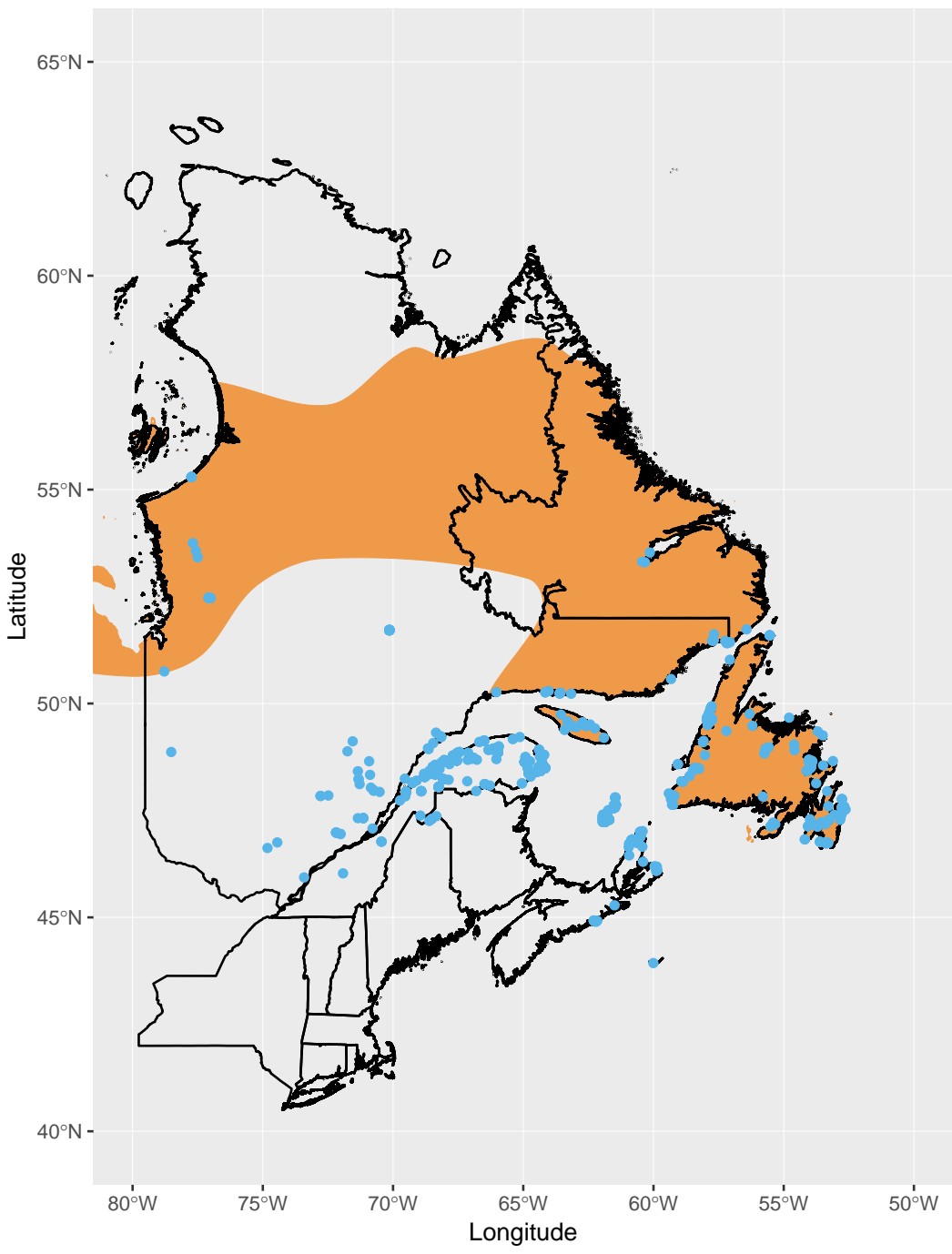

**Figure 1**  **Observations of Fox Sparrow during June and July in eastern Canada reported to eBird as of July 1990 (blue points), the approximate point at which observations of Fox Sparrows during the summer began increasing in Maine.** The putative breeding range of Fox Sparrow according to *BirdLife International and NatureServe (2014)* is shown in orange.

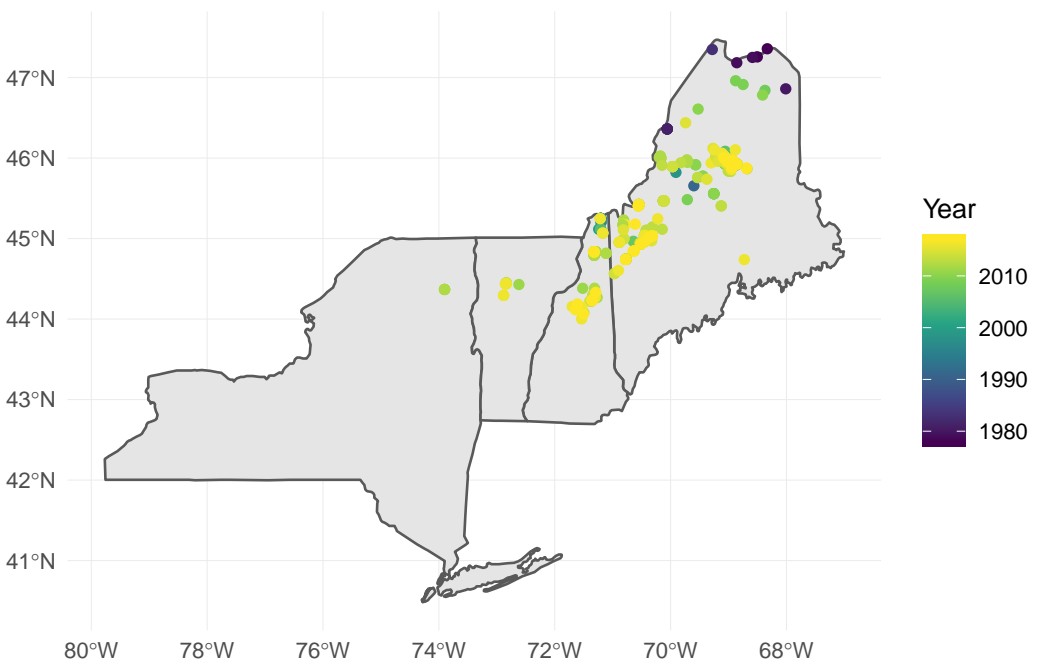

**Figure 2** Observations of Fox Sparrow during June and July in the northeastern United States reported to eBird as of July 2018.

(0/41) to a high of 8.7% (20/231 grid cells) in 2015. Despite a large increase in observer effort—ranging from a low of 145.8 h in 2005 (with a total of 20 grid cells containing at least 1 complete checklist) to a high of 3,063.5 h in 2016 (with a total of 277 grid cells containing at least 1 complete checklist)—the model containing a term for observer effort was not preferred relative to the simpler model containing only the effect of year ($P = 0.67$). Overall, the proportion of grid cells containing a complete checklist with a Fox Sparrow detection increased 18% per year from 2003–2016. Checklists containing observations of Fox Sparrow in Maine tended to occur at lower elevations (median = 480 m, range = 40 –1,592) than did checklists with observations in New Hampshire (median = 1,000 m, range = 360 –1,888) (Fig. 4).

Neither Blackpoll Warbler nor Bicknell's Thrush showed a similar pattern of increasing frequency of detection (Fig. 3). The proportion of grid cells containing a record of Blackpoll Warbler during June and July may have declined slightly, although results were equivocal ($b_{year}$ = −0.030, 95% CI [−0.064–0.005]; $P = 0.09$), whereas the proportion of grid cells containing a record of Bicknell's Thrush was apparently steady ($b_{year} = 0.03$, 95% CI [−0.029–0.109]; $P = 0.30$; Fig. 2). As with Fox Sparrow, the simpler model without an effect of observer effort was preferred for both species (Bicknell's Thrush, $P = 0.31$; Blackpoll Warbler, $P = 0.06$).

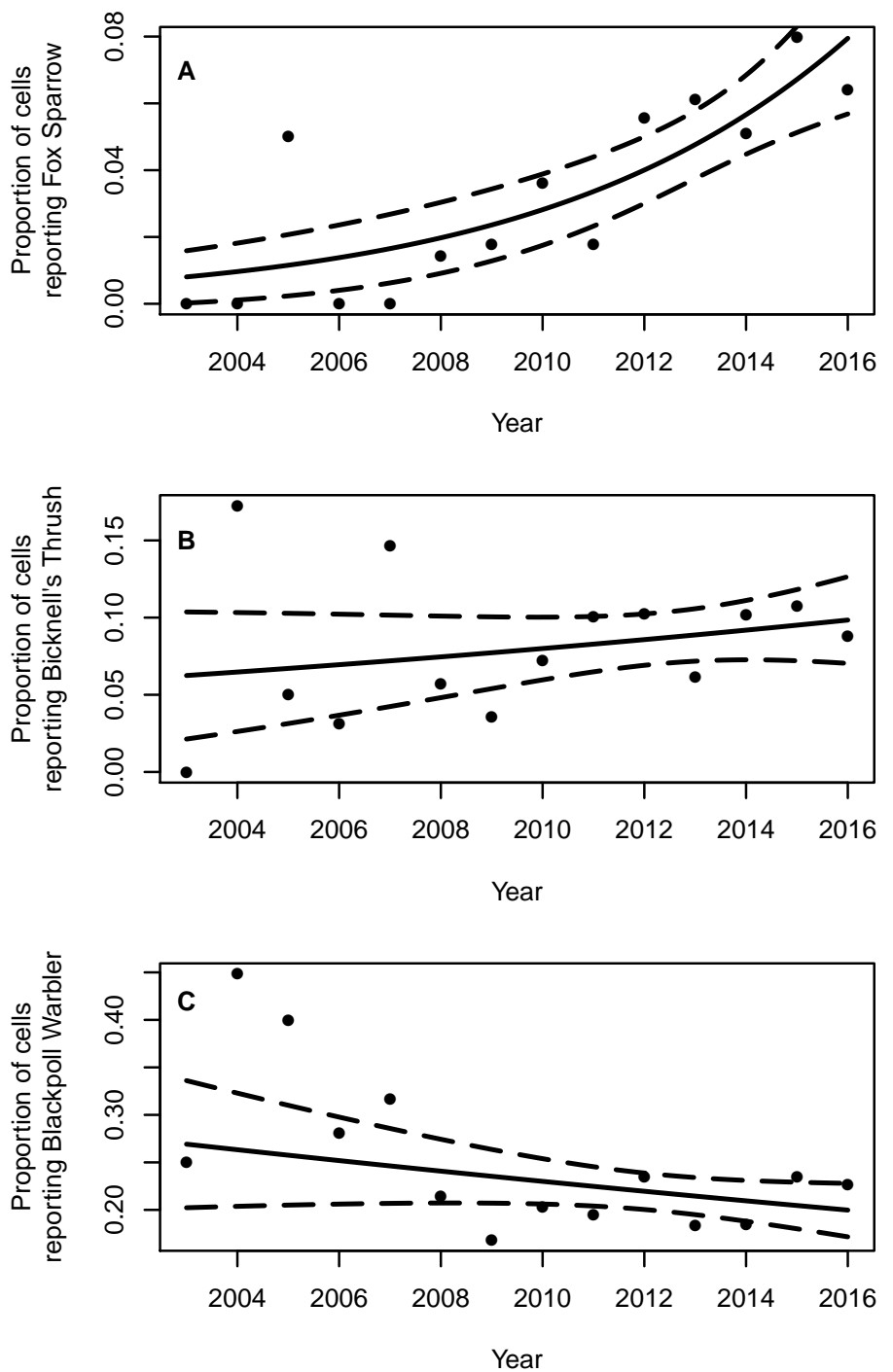

**Figure 3** **Proportion of 10 km² grid cells in northern and western Maine and northern New Hampshire containing eBird checklists with observations of Fox Sparrow, Bicknell's Thrush, and Blackpoll Warbler.** The proportion of 10 km² grid cells in northern and western Maine and northern New Hampshire containing breeding season records of Fox Sparrows reported to eBird increased 18% per year from 2003–2016 (A). Neither Bicknell's Thrush (B) nor Blackpoll Warbler (C), both of which occur in similar forest types as Fox Sparrow, showed a significant temporal trend in the proportion of grid cells reported as occupied. Solid lines show the estimated trend in frequency of grid cells with a record of the species and dotted lines indicate the 95% confidence interval around the estimated trend.

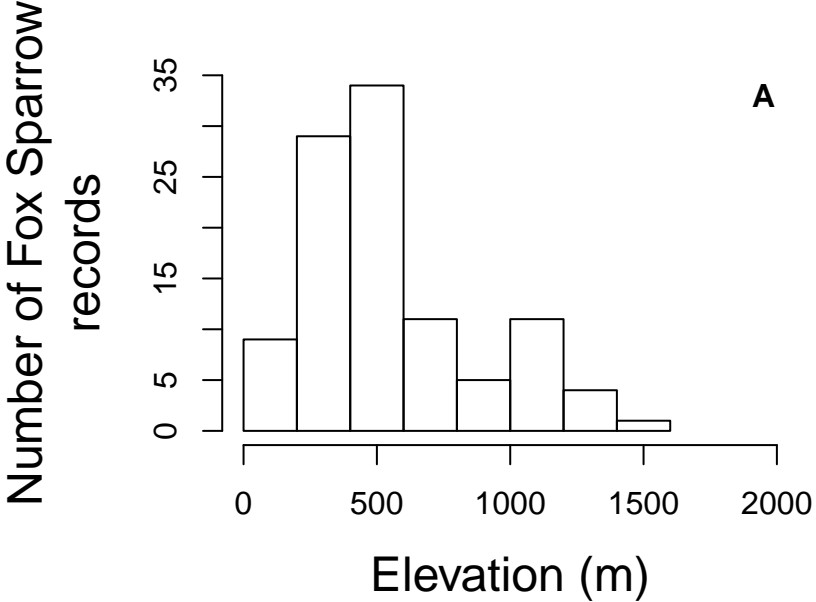

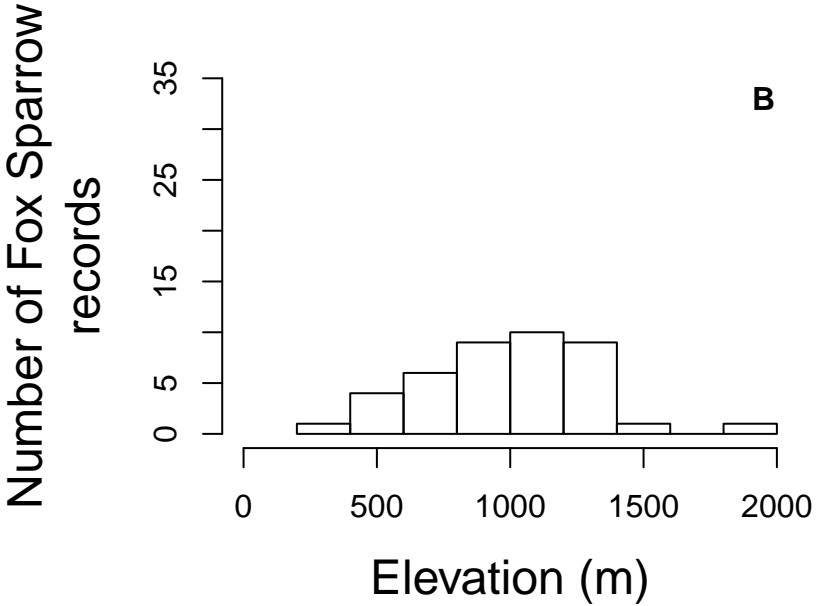

**Figure 4** **Elevation of Fox Sparrow observations on eBird checklists during June and July in Maine and New Hampshire.** Fox Sparrow observations recorded in eBird tended to occur at lower elevations in Maine (A) than in New Hampshire (B).

## DISCUSSION

Eastern Fox Sparrows appear to be in the midst of a significant southward expansion of their breeding range. I found no evidence that the species was present in the northeastern US during the breeding season until the late 1970s or early 1980s, yet they are now widely reported in both krummholz forest and in sites at lower elevations—presumably dense stands of young conifer—across New Hampshire and Maine during June and July. Observers have confirmed nesting in both states. This amounts to an approximately 400 km breeding range extension during the span of approximately 30 years.

Given that Fox Sparrows nest in remote or difficult-to-access locales—old clearcuts and mountaintops—it is difficult to rule out the possibility that they may have been present in small numbers in the northeastern US in the past. However, the avifauna of the region's mountains has been well-described since the early 1900s, yet no breeding season records for the species exist in the historical literature. Also, during the last decade, the increased frequency of breeding season eBird reports of Fox Sparrow in Maine and New Hampshire appears to reflect a real increase in the species' abundance and extent of occurrence, not an increase in observer effort in the right habitat; Bicknell's Thrush and Blackpoll Warbler, which co-occur with Fox Sparrow during the breeding season, have shown no such increase.

During the mid-1980s, birders in southern Quebec began noting what were then considered extralimital records of Fox Sparrow, at about the same time that observers in far northwestern Maine first discovered evidence that Fox Sparrows were nesting in the state. The breeding range for Fox Sparrows continued to expand south and west out of southern Quebec and northern Maine over the next decade, reaching northern New Hampshire by the mid-1990s. Although Fox Sparrows have not spread substantially further west since then—only a handful of breeding season records exist for Vermont, and only one for New York—they have continued moving south and now occur regularly during summer throughout the White Mountains of New Hampshire.

Fox Sparrows found in the northeastern US during the breeding season tend to occur either in krummholz forest at high elevations or in young stands of spruce and fir regenerating after harvest, which is consistent with descriptions of nesting habitat occupied in the core of the species' range in Canada (*Weckstein, Kroodsma & Faucett, 2002*). Other locales with potentially suitable nesting habitat include the Green Mountains of Vermont and the Adirondack Mountains of New York, both of which contain areas of krummholz forest. Extensive stands of young spruce-fir forest at lower elevations are uncommon outside of the commercial forestlands of western and northern Maine, although parts of far northeastern Vermont and northern New Hampshire support lowland spruce-fir forests that might provide suitable conditions for nesting Fox Sparrows following harvest.

Why the distribution of Fox Sparrows has expanded so rapidly is not clear. Equally puzzling is that the direction of the expansion is unexpected given contemporary climate change; in general, species have responded to rising temperatures by shifting to higher latitudes or higher elevations (*Chen et al., 2011*; *Parmesan & Yohe, 2003*). The southern range limit of 44 species of breeding birds in New York, for example, shifted northward by nearly 6 km per decade between 1980 and 2005 (*Zuckerberg, Woods & Porter, 2009*). Indeed,

models that attempt to predict the future distribution of habitat for boreal birds in general, and Fox Sparrows in particular, suggest a northward retraction of the southern range limit in response to anthropogenic climate change. For example, *Pearman et al. (2010)* and *Stralberg et al. (2015)* both predicted that by the end of the century the southern limit of the distribution of Fox Sparrow would extend no further south than central Quebec and southern Labrador. Despite the general trend for species to shift their distributions poleward and upslope in response to climate change, a substantial minority of species examined have shown the opposite pattern (*Lenoir et al., 2010*; *Parmesan & Yohe, 2003*). Explanations for range shifts downslope or towards the equator are varied, ranging from sampling error to indirect effects of climate change on biotic interactions such as competition or facilitation (*Lenoir et al., 2010*), but *Tingley et al. (2012)* argued that aspects of climate other than temperature might be a key driver of unexpected range shifts. In particular, their analysis of century-long changes in the elevational limits of montane birds in the western US suggested that whereas rising temperatures favored upslope movement of some species, increasing precipitation favored downslope movements among others. Although precipitation has not been demonstrated to produce similar effects on latitudinal distribution, precipitation across most of the northeastern US has increased by >15% over the last century (*Wuebbles et al., 2017*), which presumably could have mitigated any precipitation-related limit to the southern edge of Fox Sparrow breeding range.

Alternatively, the southward extension of the breeding range of Fox Sparrow may be unrelated to climate change. *Bateman et al. (2016)* demonstrated that the northeastern quarter of the US, from the Great Lakes region through New England, had climatic conditions putatively suitable for Fox Sparrow even though nearly all of the region was unoccupied by the species. This would suggest that either the global population was limited by factors other than habitat availability or that necessary features of the environment were absent from areas that were climatically suitable. To the latter point, Breeding Bird Survey data reveal no evidence of population increases in eastern Canada (*Environment and Climate Change Canada, 2017*), as might be expected if growth in the core of the range was facilitating the southward expansion of the breeding range.

However, the expansion of breeding Fox Sparrows into Maine coincides approximately with the extensive salvage logging that occurred during and after the last outbreak of eastern spruce budworm, which created vast areas of young spruce-fir forest in northern and western parts of the state. The epidemic began in the 1960s and continued through the 1980s, and was accompanied by extensive salvage harvests from the 1970s into the 1990s (*McWilliams et al., 2005*). As regeneration proceeded, the area of young spruce-fir forest expanded significantly, from $< 4,000$ km$^2$ in 1982 to nearly 10,000 km$^2$ by 2013, with nearly all of the increase occurring between 1982 and 1995 (*McCaskill et al., 2016*). Furthermore, the increase in the area of young spruce-fir forest was greatest in the northern counties of Aroostock, Somerset, and Piscataquis, where breeding Fox Sparrows were first detected (*McWilliams et al., 2005*). Given that Fox Sparrows will nest in young conifer stands (*Weckstein, Kroodsma & Faucett, 2002*), the species' southward expansion may have been in part due to the increased availability of nesting habitat created by logging; under this scenario, climatic conditions for breeding Fox Sparrows may have always been suitable

in the northeastern US, but the lack of dense, short-statured stands of conifers limited their ability to successfully nest in the region. *Banks (1970)* suggested a similar role for logging in the spread of Fox Sparrows onto the western slopes of the Cascade Mountains of Oregon, where a combination of clearcutting and burning created extensive new areas of shrubby, early-seral vegetation in an area that was previously dominated by mature coniferous forests. Increased availability of stands of young conifer may also explain the higher incidence of observations of Fox Sparrow at low elevations in Maine. Maine has substantially greater acreages of young balsam fir (*Butler, 2017*; *Morin & Lombard, 2017*) and for practical and regulatory reasons nearly all of the timber harvest that creates young fir forest in Maine occurs at elevations <800 m, which corresponds to the peak in observations of Fox Sparrow evident between 250 and 750 m. However, the role of early-successional forests in promoting the range expansion documented here is speculative given the lack of data on habitat conditions at any of the occurrence locations at lower elevations. Targeted surveys across a range of forest types would likely prove useful in clarifying the connection between young forest and breeding Fox Sparrows in the northeastern US.

Fox Sparrows are unusual in exhibiting a southern expansion of their breeding range at a time when most bird species of the northeastern US appear to be shifting poleward in response to climate change, but they are not unique. *Hitch & Leberg (2007)* found that, on average, the southern range limit of birds of the northeastern US during 1998–2002 was the same as it was from 1967–1971, but also reported that 6 species had shown significant southward range extensions over those intervals. One, Bobolink (*Dolichonyx oryzivorous*), is a grassland species, and two, Pine Siskin (*Carduelis pinus*) and White-winged Crossbill (*Loxia luecoptera*), are highly nomadic granivores. Yellow-rumped Warblers (*Setophaga coronata*) extended their range southward, perhaps in response to the recovery and replanting of conifer forests (*Hunt & Flashpoler, 1998*). Distributional shifts in the other two species that *Hitch & Leberg (2007)* reported to have moved southward, Alder Flycatcher (*Empidonax alnorum*) and Mourning Warbler (*Geothlypis philadelphia*), have not been carefully studied, but notably both species use early-successional habitats and, like Fox Sparrow, may have benefited from increased disturbance of mature forest (*Pitocchelli, 2011*; *Lowther, 1999*). However, *Hitch & Leberg (2007)* estimated that all three of these species expanded their breeding range southward by <100 km, far less than the approximately 400 km extension exhibited by Fox Sparrows in this analysis.

Although this study did not address changes in the northern limit of the breeding range of Fox Sparrow, some climate models predict a northward expansion of Fox Sparrows into the arctic as shrub cover increases in arctic tundra (*Thompson et al., 2016*) even as the southern range limit contracts. Empirical observations appear to confirm this prediction; *Whitaker (2017)* reported, based on surveys conducted from 2008–2016, that Fox Sparrows were breeding up to 100 km north of their historic (i.e., early- to mid-20th century) northern range limit in Labrador, and he attributed this range extension to increased shrub cover associated with climate change. Taken in tandem with the data presented here, a picture emerges of Eastern Fox Sparrow breeding range expanding significantly to both the north and south during the past several decades.

## CONCLUSIONS

Eastern Fox Sparrows were historically a passage migrant through the northeastern US during late spring and early fall, but within the past few decades have become widespread and regular during the breeding season. Although confirmed nesting records remain scarce, the consistent presence of singing males suggests that the southern limit of the breeding range now extends several hundred kilometers south of where it is placed on most published range maps. In particular, the species' breeding range now encompasses the northwestern half of Maine and the highlands of northern New Hampshire from the White Mountains to the US-Canada border. Scattered recent records in Vermont and New York, along with the presence of suitable habitat in the higher mountains of both states, suggests that Fox Sparrow may continue expanding its range to the west.

## ACKNOWLEDGEMENTS

Thanks to CC Rimmer and KP McFarland for constructive reviews of an early draft of this manuscript.

### Funding

The authors received no funding for this work.

### Competing Interests

The author declares that they have no competing interests. John D. Lloyd is an employee of Vermont Center for Ecostudies, a 501(c)(3) non-profit corporation.

### Author Contributions

- John D. Lloyd conceived and designed the experiments, performed the experiments, analyzed the data, contributed reagents/materials/analysis tools, prepared figures and/or tables, authored or reviewed drafts of the paper, approved the final draft.

### Data Availability

Lloyd, John (2018): eBird checklist data used to analyze temporal patterns of occurrence of Fox Sparrows in the northeastern United States. figshare. Dataset. https://doi.org/10.6084/m9.figshare.7283144.

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
