# Peer review of "The recent expansion of Fox Sparrow (Passerella iliaca iliaca) breeding range into the northeastern United States"

_PeerJ, doi:10.7717/peerj.6087_

## Round 0.1 · original submission · Minor Revisions

Three reviewers believe your paper is relevant, well organized, and fits well with PeerJ’s goals. I agree with them. However, there some minor changes that need to be made in the manuscript before it can be accepted for publication. When resubmitting your manuscript, please carefully consider ALL points mentioned in the reviewers' comments, explain every change made, and provide proper rebuttals for any remarks not addressed. I look forward to seeing an updated version of your paper.

# ·

Basic reporting

This manuscript is well written and well analyzed. The figure works.

Experimental design

The author has done an adequate job of gathering data to demonstrate the expansion of the distribution of Fox Sparrow to the south in the eastern portion of its range.

Validity of the findings

These findings are quite valid and interesting to ornithologists

Additional comments

With respect to the discussion, no mention is made of the fact that expansions southward are somewhat counter-intuitive. This can be seen in elevational studies to (e.g., Mortiiz et al. 2008, see Montane Shrew). There is also nothing about historical range change in this species which over the last 10,000 years probably included many of the areas of the recent expansion. I think the author could also bring up the issues of climate change and what was predicted for Fox Sparrow (and possibly examining Blackpoll Warbler and Bicknell's Thrush) by recent broad analyses of North American birds that have suggested climate will drive most species northward.

Here are couple of other recent studies on avian range expansion in North America. I think putting what Fox Sparrows are doing in the broader ornithological perspective is too good an opportunity to miss.

Ingenloff, K., Hensz, C.M., Anamza, T., Barve, V., Campbell, L.P., Cooper, J.C., Komp, E., Jimenez, L., Olson, K.V., Osorio-Olvera, L. and Owens, H.L., 2017. Predictable invasion dynamics in North American populations of the Eurasian collared dove Streptopelia decaocto. Proc. R. Soc. B, 284(1862), p.20171157.

Butcher, J.A., Collier, B.A., Silvy, N.J., Roberson, J.A., Mason, C.D. and Peterson, M.J., 2014. Spatial and temporal patterns of range expansion of white‐winged doves in the USA from 1979 to 2007. Journal of biogeography, 41(10), pp.1947-1956.

Specific comments:

Line 160. This is a little bit of an overstatement. While there is documentation that the species occurs on the high mountains of the Adirondacks. This is based on a single record (so far).

Line 198. yet they are now widely reported in New Hampshire and Maine during June and July. This isn't really true. It is just a wording issue, but Fox sparrows are now reported from the high elevations in coniferous forest of northern and western Maine and northern New Hampshire.

Line 232. Change to: "Exapansion of breeding Fox sparrows"

Line 236. "Banks (1970) suggested a similar explanation for the spread of Fox Sparrows onto the western slopes of the Cascade Mountains of Oregon (but see Marshall et al. (2003))."

This should be expanded on, clearly Marshall disagreed with this hypothesis, why?

Figure 3. I think this could be made better by adding in data that addresses the availability of suitable habitat across elevations in polygons related to the area of expansion. I assume the high frequency of lower elevation records in Maine could be related to latitude and/or habitat management.

Reviewer 2 ·

Basic reporting

The author has submitted a well written manuscript that adheres to the expected structure of a scientific contribution. The author has consulted a broad array of appropriate literature and citizen science data. I was particularly pleased to see that the North American Birds series had been consulted, though I was surprised that no citations came from the review of the regional summaries; surely those authors have also pointed out the increase in this species? The figures are clear, and the first two figures very clearly support the Discussion. I am on the fence about whether Figure 3 is needed, as I suspect that text including the mean and standard deviation of these sightings might be sufficient (there is not much text otherwise spent describing or interpreting this pattern in the Discussion).

I believe that this manuscript is missing the required dataset submission. The author indicates that all data are publicly available through eBird. However, additional records may come to light as existing, older field notes are added to the database, or as other records are re-reviewed when regional reviewers adjust filters, and this could potentially change the results of a repeated analysis. As it stands now, the reader cannot completely duplicate the analysis that the author undertook. I would suggest following the PRISM workflow that PeerJ requests for systematic reviews, and then upload the Excel files from the resulting eBird datasets.

Did the author check The Kingbird journal for additional New York records?

Experimental design

The author has focused on a regional expansion of a summering species into the northeastern United States. While this paper presents no new data, it is a thorough review of published records that have not been assembled into a single narrative before. Birders and ornithologists in the region are already aware of the expanding range of this species as a breeder, but the expansion has not been thoroughly reviewed before. The extra effort made to compare frequency, standardized against two more common species, is appreciated.

My concern, however, relates to the PeerJ expectations for this kind of manuscript. PeerJ states, "Manuscripts that report range extensions, life history information, or the description of new species or other taxa should address a biological question or hypothesis as the focus of the submission." While this submission does place the story of range expansion into the context of a hypothesized influence from increased habitat modification in the region, the manuscript does not explicitly test this. To do this, the author would need to quantify which areas have increased pressure and compare this to observed increases in breeding season records. I am not sure that this level of forestry data is available, so I wonder if there could be a rougher scale approach comparing areas with no logging pressure versus those with pressure (e.g. compare wilderness to non-wilderness mountains within State and National Forests)?

The author cites a few instances of documented nesting, but most of these records do not have documentation of nesting. How many of these observations were of two individuals, and are there any data on females being present? Or are they all unpaired, singing males? There is a difference between unpaired males summering south of their range versus breeding there. Both are part of a single phenomenon, I would think; as a species expands, the vanguard would be unpaired males. But if almost all of these observations are of unpaired males, then this manuscript is documenting a changing behavior of unpaired (and presumably second calendar year individuals) rather than a difference in breeding distribution. We see this in lots of other species, where younger birds don't make it to the core breeding range for their first summer as an adult (particularly in shorebirds, but there are many, many reports of passerines doing this in the pages of North American Birds and in eBird).

Validity of the findings

This manuscript demonstrates sufficient scientific rigor to accomplish its goal of documenting the increasing presence of Fox Sparrow in New England in summer.

The comparison of the expanding range to the spruce fir regional harvest rates needs more investigation and quantification. This is related to the hypothesis testing issue I listed in Experimental Design, is critical for the context of this expansion. No references are cited to back up any statements about increased or changing forestry regimes. This is particularly needed in lines 233-234.

Line 97 indicates that the choice of resolution for the grid was arbitrary. I agree that there is no true guideline for size to use, but I would like to see investigation of the effect of grid size on the results. 10 km^2 strikes me as being quite large for this study; does revisiting with 5 and 1 km^2 grids change your results? Your overall study covers very few grid cells at the 10 km^2 resolution.

Additional comments

I have a minor note about scientific/common names here. The Red Fox Sparrow is an informally recognized common name (common names in North America are only assigned at the species level by the American Ornithological Society's Check-list Committee). In general usage, Red Fox Sparrow refers to a subspecies group, not a particular subspecies. That is, Red Fox Sparrow is P. i. iliaca and P. i. zaboria. This contribution focuses on P. i. iliaca, and the common name of Eastern Fox Sparrow is typically assigned to this one (see Rising 1996, for example). The manuscript's title is correct, but there are some additional uses within the text that should be corrected, particularly line 30.

eBird is cited incorrectly; they prefer that you use this: Sullivan, B.L., C.L. Wood, M.J. Iliff, R.E. Bonney, D. Fink, and S. Kelling. 2009. eBird: a citizen-based bird observation network in the biological sciences. Biological Conservation 142: 2282-2292. I would also suggest adding some text to explain that these are vetted observations by peers, not just acceptance of all observations from observers at all experience levels.

Need a broader citation for North American Birds as a journal (list publisher, at minimum) and explain that the other titles are previous titles for the same journal. Maybe a phrase to explain that this is a vetted review of regional bird records for each season?

Line 57: delete hyphen in "breeding-bird"

Line 90-92: awkward sentence, could be rephrased or broken into two to explain why this decision was made.

Lines 154, 165, and elsewhere: can remove the minus in front of the longitude, as indicating that the degrees are West makes the minus unnecessary.

Line 167: extra parenthesis can be deleted.

Line 200: delete hyphen in "breeding-range"

Lines 204, 208, 217: delete hyphen in "breeding-season"

Lines 206-207: need to include some citations to back this up, I'm not sure which of these efforts would have been focused widely enough to describe rarities as opposed to the budworm-focused species.

Line 212: need a citation for the Quebec trend.

Figure 2: delete hyphen in "breeding-season". Figure legend should indicated what the solid and hyphenated lines mean.

·

Basic reporting

This manuscript is well put together, well written and I see no issues with the general outline of the manuscript.

Experimental design

This is not an experiment as such, but an analysis of a set of observations in the wild, so this feature not completely relevant. However, it does in general meet the criteria for ths section. The issue it analyzes is interesting. The methods used for analysis make sense and are described in good and clear detail.

Validity of the findings

The finding seem to be valid and the comparisons made to other species make sense and strengthen the argument that this change in Fox Sparrow distribution is real and not a reflection of increased observer activity in the region or habitat. I personally would like to see more speculation or consideration of what this distributional change indicates by considering other bird species found in the same habitat in this range and other species that are expanding their range southward in this part of North America.

Additional comments

My main concern with this paper is that there is not a real attempt to place the changes in distribution of Fox Sparrow into context with other species in the region. The two groups of species, I’d like to see some consideration of would be species that are extending their breeding range south in the NE US, and those species that occupy the same sort of habitat as Fox Sparrow. In terms of species extending their range, I don’t know the specifics of the part of North America, but I know for example has Merlin has extended its breeding range south across much of its range in eastern North America. Are there others, perhaps species more focused in NE US, that might give insights into the changes happening in Fox Sparrow. Similarly, Fox Sparrow shares its habitat with some other species. I’m thinking particularly of species that occupy early successional clearings in coniferous forests. While Bicknell’s Thrush and Blackpoll Warbler are, I think, good species in terms of range and association with coniferous forest, neither are really tied to the specific habitat that Fox Sparrow uses. Gain, I don’t know the details of bird distributions in the northeast well enough to suggest a suite of species. The only one that occurs to me is Mourning Warbler. It has a much broader range and I wouldn’t expect them to be expanding their range, but you might see evidence of greater abundance or habitat occupancy if the Fox Sparrow response is not unique to that species. I don’t really expect a detailed analysis of other species, but at least some consideration of other species with analogous range changes or occupying habitat that might be responding to changes in forestry management in similar ways.

It would be really helpful to have a map that shows the range (or at least southern limit) of breeding Fox Sparrows in eastern Canada at some reference time period when the expansion begin. This would put the expansion into the NE US more in context

Line 100 People entering traveling counts into e-bird are supposed to place their point at the midpoint of the route they traveled, not the starting point as stated here. Doesn’t really change this issue, but might reduce the number of times that an observation came from the “wrong” grid cell, since the farthest the observer birded from the mapped point would be half the distance it would be if the starting point was mapped.

Line 129 States that Keith and Fox 2013 describe Fox Sparrow as common migrant in New Hampshire, but line 154 gives Keith and Fox as source for breeding record from New Hampshire. I think the seasonal status of Fox Sparrow in New Hampshire given by Keith and Fox needs to be better described.

Line 151-152 This sentence provides information on occurrences in Maine in 1999 and 2000, but reference supporting this is a 1999 paper, which couldn’t possibly have information on a 2000 year.

Line 167-169 This is a run on sentence that should probably be broken up. I would put a period after “significance” and delete “but.” Also I think the ) following July 2016 should be a comma.

Lines 167-174 Can any sources be cited for the information in this paragraph?

Lines 186-188 The difference in elevational distribution between Maine and New Hampshire is interesting and a figure (Figure 3) is used to support this difference. In the discussion, this difference is not talked about at all. The paragraph of lines 220-228 suggests to me that the lower elevation distribution in Maine and the fact that the summering records in Vermont and New York are in high elevation sites reflects habitat availability and not some other factor. However, the paragraph at 220-228 does not mention the elevational distribution of the summer records at all. One alternate possibility I have considered is that it is a reflection of time that the population has been established. The idea here is that Fox Sparrows expands range into high elevation krummholz habitat, and as the population expands is able to move into the structurally similar lowland early successional areas in logged spruce forest. I wonder if there would be any temporal pattern in the elevational distribution of birds in Maine and New Hampshire with earlier records being concentrated at high elevations.

Lines 204-207 Can any sources be provided for “ornithological investigations” in the 1980s in this geography mentioned in this sentence.

Line 236 – 238 I think it would be worth an additional line or 2 about the Oregon situation because the suggestion of Marshall et al is that the extension of the known range to the western Cascades might be a reflection of accessability rather than an actual range change. Given that they don’t provide any data, this is purely speculation, and given that this paper goes to some trouble to deal with this possibility in the NE US, I think it is worth being more explicit here about the parallels between the two cases.

---

## Round 0.2 · accepted · Accept

I have checked all the changes made in your paper as well as the rebuttal letter. You addressed all the major suggestions made by the reviewers, so I consider your article as ready for publication. Please work with our production team. Congratulations!

#